# mm-DSF: A Method for Identifying Dangerous Driving Behaviors Based on the Lateral Fusion of Micro-Doppler Features Combined

**DOI:** 10.3390/s22228929

**Published:** 2022-11-18

**Authors:** Zhanjun Hao, Zepei Li, Xiaochao Dang, Zhongyu Ma, Yue Wang

**Affiliations:** 1College of Computer Science and Engineering, Northwest Normal University, Lanzhou 730070, China; 2Gansu Province Internet of Things Engineering Research Center, Lanzhou 730070, China

**Keywords:** hazardous driving behavior, FMCW radar, mm-DSF, micro-doppler spectrum, data lateral fusion, dual-stream branching network

## Abstract

To address the dangerous driving behaviors prevalent among current car drivers, it is necessary to provide real-time, accurate warning and correction of driver’s driving behaviors in a small, movable, and enclosed space. In this paper, we propose a method for detecting dangerous behaviors based on frequency-modulated continuous-wave radar (mm-DSF). The highly packaged millimeter-wave radar chip has good in-vehicle emotion recognition capability. The acquired millimeter-wave differential frequency signal is Fourier-transformed to obtain the intermediate frequency signal. The physiological decomposition of the local micro-Doppler feature spectrum of the target action is then used as the eigenvalue. Matrix signal intensity and clutter filtering are performed by analyzing the signal echo model of the input channel. The signal classification is based on the estimation and variety of the feature vectors of the target key actions using a modified and optimized level fusion method of the SlowFast dual-channel network. Nine typical risky driving behaviors were set up by the Dula Hazard Questionnaire and TEIQue-SF, and the accuracy of the classification results of the self-built dataset was analyzed to verify the high robustness of the method. The recognition accuracy of this method increased by 1.97% compared with the traditional method.

## 1. Introduction

The research and development of motion recognition technology have facilitated the development and application of wireless communication in cross-scene conditions. At the same time, the glory of dangerous driving behaviors has attracted the attention of researchers due to its specificity. Destructive driving behavior is the driver’s operation or illegal interaction with others in violation of traffic safety regulations during the driving process, resulting in potential safety hazards for people and vehicles, such as voice-connected navigation, answering phone calls, and turning the body with hands off the steering wheel during driving. The current mainstream wireless sensing action recognition technology is the method of received signal strength [1] (RSS) and based on channel state information [2] (CSI), as well as wearing traditional wearable sensors on the human body to detect [3] the target local action can achieve a better recognition effect. Still, in the driving environment, wearable sensors will make the wearer produce a particular action hindrance. Wi-Fi technology has made its mark in indoor sensing applications due to its universality and the fact that it does not need to be worn [4]. Motion recognition in the cab, however, is distinguished from traditional technologies by conditions that limit the small space, the density of people, and greater accuracy [5]. To accurately perceive the subtle movements of the target person and identify the target motion of the signal-enclosed carrier, there are three typical techniques. The first is LIDAR, commonly known as four-line LIDAR Ibeo, and 3D LIDAR Velodyne, which uses linear interpolation based on the XYZ axis point data collected by LIDAR, using an obstacle occupancy grid map and a search continuum algorithm. It has been widely used in vehicle obstacle avoidance systems in transportation [6]. The second is sensor sensing technology, such as IMU sensors, to sense the surrounding environment and collect important information such as visual images of the target. Still, the cost is high, and the image information data collected in the vehicle are subject to loss of information such as target shape, shadow, and diffuse ditch reflection, and are prone to receive interference from the light intensity index [7]. The third is this paper uses the millimeter-wave passive sensing technique. FM continuous wave is characterized by a high sensing accuracy, and the waveform signal is less affected by the environment [8,9,10,11,12,13,14,15]. Due to the unreliable deployment of Wi-Fi in the vehicle and since the driver’s position is fixed, the choice of direction-sensitive millimeter-wave radar can provide high-accuracy identification and a low error rate. 

Conventional LIDARs have better angular resolution than millimeter waves but require software platforms and back-end servers, which are expensive and make the commercial cost challenging. In the literature [16], Dong et al. proposed FM continuous wave-based gesture recognition by introducing distance, azimuth, and elevation information for HGR processing, with 3D-FFT jointly encoded multidimensional features through mutual relationships, showing good performance in solving enhanced target action representation. In [17], Zhang et al. presented an FM continuous wave radar tracking pedestrian trajectory detection system (MRPT) mounted outside the vehicle using target presence probability and Markov transition matrix integration for continuous pedestrian detection with a low signal-to-noise ratio, showing good performance. With innovations in detection technology and privacy intrusion issues, drivers do not want to capture RGB image video information in private spaces, and the visual approach is susceptible to lighting conditions inside the cab. In [18,19], a visual image approach is used to detect inattentive driving behavior (HIDB), specifically subdivided into distracted (DD), fatigue (DF), or drowsiness (DFD), using graph neural networks and Markov models in the SlateFarm dataset to discuss the factors that influence driving behavior.

Millimeter-wave radar can obtain the radial distance of a target by transmitting an FM continuous signal, and distance-Doppler motion detection is achieved by a matrix composed of Chirp [20]. Denoising also provides detection thresholds, and the radar performs constant false alarm detection (CFAR) in signal processing, using cell values to compare whether the detection cell is a target [21] as with Wi-Fi it is more difficult to extract features in the complex conditions of the spatial environment. In this experiment, we propose an in-vehicle hazardous driving action recognition based on WiGig [22] technology, which extracts micro-Doppler features as feature values through radar signal processing, divides the self-built dataset into two channels into an mm-DSF CNN network according to data sparsity, and matches the collected actions with the training set to achieve an early-warning effect [23,24]. In this paper, we use an excellent combination of micro-Doppler features collected by radar to accurately sense subtle motions. The feature information is fed into an improved mm-DSF that significantly mitigates the undesirable effects of dangerous driving behavior by classifying high-density matrix data in the vehicle using dual-channel point-to-point data density due to differences in the local feature dimension of the target and the large amount of collected data.

## 2. Radar Data Processing

### 2.1. Radar Implementation Method

Millimeter-wave radar is an integrated single-chip processor based on FMCW (frequency-modulated continuous wave) radar scanning technology. This paper uses a large bandwidth of 4 GHz high-frequency linear constant pulse that can significantly improve the efficiency of acquiring human echoes. To obtain the optimal accuracy of the perceived bias angle and Doppler distance for the driver in complex scenes in the car, the basic parameters of the radar are set as follows: transmit power 12.5 dBm, maximum FM slope 100 MHz/us, IF bandwidth 175 KHz~5 MHz, and maximum ADC sampling rate 12.5 Msps (real)/6.25 Msps (complex). The transmitting antenna emits a high-frequency carrier wave, indicating:(1)$$S_{T}(t)=A_{T}\cos\left[2\pi f_{c}t+2\pi \int_{0}^{t}f_{T}(\tau )d\tau \right]$$
where *T* is the period of the high-speed FM continuous wave, $T_{c}$ is the pulse width of the linear FM wave, $A_{T}$ is the amplitude of the transmitted signal, τ is the time delay between the transmitted signal and the return signal, *S* is the slope of the FM transmitted wave of the FMCW radar, *c* is the value of the speed of light, and *d* is a differential operator.

The time node at which the FM continuous wave is transmitted to the moment at which the waveform is received produces a time delay and a Doppler shift due to the movement of the driver. With the model of the transmitted frequency signal, the signal received by the receiving antenna is considered to change in the frequency domain and generate a time delay after Fourier transforms, so that the received signal can be derived as follows:(2)$$S_{R}(t)=A_{R}\cos\left[2\pi f_{c}(t-\Delta t)+2\pi \int_{0}^{t}f_{R}(\tau )d\tau \right]$$
where $A_{R}$ indicates the amplitude of the received signal. The first term in the expression is ignored because the time delay is very small due to the extremely fast propagation of electromagnetic waves. $f_{IF}$ is only related to the number of sampling points, x, in the target FM period. The IWR1642 radar system used in this method includes a transmitting component, TX, and a receiving component, RX, an RF component and clock timing simulators, TF-RF and RX-RF, an analog-to-digital converter (ADC), microcontroller (MCU), and a digital signal processor (DSP). The IF signal processed by blending is connected to the user’s experimental terminal through the port of the DCA1000EVM data acquisition card. The acquired signals are digitally parsed. The parsing process is performed to store the acquired echo information of the six ADC channels in the form of an ADC_data.bin file into the internal memory with ECC and reorganize them into a data matrix according to the number of antennas of the received channels. The data length of the received media can be parsed by radar. Figure 1 illustrates the entire signal processing process of the driver’s echo signal into the radar.

### 2.2. FM Continuous Wave Echo Monitoring

Radar basic data sensing can generate dynamic features in three dimensions, including distance, velocity, and angle information. One of the distance features used to locate the critical position of the target uses the principle that the distance measurement of the radar system to the target in the vehicle is based on the round-trip propagation time difference of the electromagnetic wave signal between the antenna and the driver. The instantaneous distance between the radar and the target for a given time window can be expressed as the following equation:(3)$$R=\frac{c\cdot \Delta t_{d}}{2}$$

In the millimeter-wave radar system, the delay of $\Delta t_{d}$ is very short, and the delay of echo signals at different distances will have slight differences. The instantaneous frequency of the two signals synthesized into the intermediate frequency signal is the phase difference between the two input signal waveforms, so the frequency of the mixed IF signal is used to represent the transmission delay, $\Delta t_{d}$, and the intermediate frequency signal can estimate the distance to the target. Therefore, the phase of the output signal, $X_{out}$, can be expressed by the following equation:(4)$$X_{out}=\sin\left[\left(\omega_{1}-\omega_{2}\right)t+(\Phi_{1}-\Phi_{2})\right]$$

When faced with a single identified target, the chirps of TX and RX as a function of time can be represented as a visualization showing the time–frequency of the IF signal, where the chirps of the transmit and receive antennas are parallel. The method uses the device’s characteristic principle, whereby the distance between two lines is the single-tone signal value. The region where the waveform exists is limited to the overlapping timing of the TX sending the FM pulse and the RX receiving the FM pulse, for example, as shown in Figure 2. The initial phase of IF ($\Phi_{0}$) is where the IF signal begins, and the phase difference between the FM pulses sent by the transmit antenna and the FM pulses received by the receive antenna can also be expressed by the equation:(5)$$\Phi_{0}=2\pi f_{c}\tau =\frac{4\pi d}{\lambda }$$

The AOA angular signature is based on the fact that small changes in the distance from the target to the transmitter result in a phase change in the peak of the distance—fast Fourier transform or Doppler-fast Fourier transform. The different Δds paces from the target to each antenna result in a phase change of the FFT peak, which is used to estimate the angle of arrival. Assuming that the separation distance between the receiving antennas is *d*, the rise of the arrival is calculated by the following equation from the Pythagorean theorem, Δd=lsin(θ):(6)$$\theta =\sin^{-1}(\frac{\lambda \Delta \phi }{2\pi d})$$

Velocity has a range interval to avoid velocity blur, and the angle likewise has a break to prevent the field of view. The maximum angular field of view of the radar is determined by the maximum AOA that can be estimated. When Δϕ>π, this creates an angular ambiguity to the extent that it is impossible to decide on the target’s orientation:(7)$$\Delta \phi =\frac{2\pi d\sin(\theta )}{\lambda }<\pi \Rightarrow \theta_{\max}=\sin^{-1}(\frac{\lambda }{2d})$$

## 3. Action Classification Models and Identification Methods

### 3.1. Overall Model Flow

This paper proposes a two-channel deep mm-DSF model based on micro-Doppler features to address the needs and limitations of the driver’s dangerous action analysis inside the vehicle. First, the millimeter-wave radar is placed in the monitorable area inside the car under test. The micro-Doppler spectrum of the dangerous action is obtained from the frequency change of the time–frequency difference signal as well as the FM signal through baseband processing and fast Fourier transform. The spectrum information is integrated and input into the method according to the action amplitude. It is divided into a two-channel input model. The micro-Doppler features accurately classify the dangerous action. The overall process of the model is shown in Figure 3.

### 3.2. Doppler Feature Vector Extraction

In this research experiment, each frame contains 128 chirp signals, for which the chirp signals can be reflected and mixed by a mixer to obtain the intermediate frequency signal (IF). Each chirp signal can be manually set up with 128 sampling points for the experiment. When representing each frame in terms of chirp signals, a matrix of 128 × 128 is used to describe the structures, which are formatted in row×column numbers based on the row numbering of the IF signal. A Doppler FFT is performed on the data sampling points in the frequency band of the intercepted chirp signal during the frame period of the target action signal, yielding a distance Doppler distribution based on driving hazard maneuvers. A distance Doppler spectrum can be obtained for a 1D chirp signal by a fast discrete Fourier transform.

In contrast, a distance–velocity Doppler spectrum can be obtained for a 2D chirp signal by a fast discrete Fourier transform. The acquired distance Doppler spectrum distribution will produce a constant false alarm rate (CFAR), not only for individual target detection but also for separating uncorrelated target reflections and handling thermal noise of various intensities in signal reception and indexing the velocity of individual targets to obtain 2DFFT results for the acquired bending over in driving to pick something up. As an example, with the x-axis as distance, the y-axis as velocity, and the z-axis as signal strength, from the projection to the horizontal plane corresponding to the peak distance spectrum shown in Figure 4a, the distance and speed information spectrum of the target dangerous driving detection in Figure 4b can be obtained. For multiple targets in the vehicle, CFAR detection processing is also required.

According to the physical characteristics of the human body, the detection part of the target driver will reflect and inevitably scatter the signal sent by the transmitter, so the number of echo information frequencies received by the receiver varies relatively. There are different offsets of target velocity during the process of Puller information. The Doppler shift is invariant over the fixed FM period of the signal, and Δ*f_d_* is related to the index number *y* within the limited FM period. The entire acquisition process occurs during the “inter-frame” period of idle time between the adequate linear FM time and the end of the frame. Therefore, the IF signal of the self-built dataset is mixed with the FM signal in a frame period and can be expressed as the following equation, discretely:(8)$$s_{IF}\left(x,y\right)=A_{IF}\left(x,y\right)\cdot \exp\left\{j2\pi \left[f_{IF}\left(x\right)-\Delta f_{d}\left(y\right)\right]x/F_{s}\right\}$$where $A_{IF}\left(x,y\right)$ denotes the amplitude of the IF signal, and $F_{s}$ indicates the sampling rate of the ADC analog to electrical conversion. Fourier one-dimensional transformation of the sampled chirp signal is distributed according to the time domain characteristics in any FM period, *y*, of the acquired frame. Obtain the frequency of the transmitted signal about the return IF signal, $f_{IF}\left(x\right)\left(x=1,2,3,\cdots ,N_{adc}\right)$, or the velocity-dependent Doppler shift of the Fourier 2D transform of any chirp signal period sequence in the frame $\Delta f_{d}\left(y\right)\left(y=1,2,3,\cdots ,N_{chirp}\right)$, where $N_{adc}$ is the number of samples per FM cycle, and $N_{chirp}$ is the FM signal period per frame. The feature vectors were extracted from the micro-Doppler spectrograms for each maneuver, and Figure 5 shows the left-turn 90° micro-Doppler features for the nine hazard maneuvers in the dataset.

### 3.3. Movement Analysis and Classification

#### 3.3.1. Rectangular Action Area Characterization Detection

In the acquisition of hazardous driving data features, the effective radar coverage area in the vehicle is defined by the distance and azimuth data, and the range of values of the detection area is independent of the target and related to the installation position and azimuth attitude of the radar signal transmitter/receiver. The experimental objective is to define the driver’s seat position as the target area for signal acquisition, and the driver’s target area defined by the distance–azimuth expression is expressed by the following inequality:(9)$$Z_{i}=\left(n,m\right):r_{l}^{j}\leq r_{n}<r_{u}^{j}$$
(10)$$\theta_{l}^{i}\leq \theta_{m}<\theta_{u}^{i}$$
where $Z_{i}$ is the area number in the set of all distance–azimuth angles within the rectangular boundary of the target area, the parameter variables $r_{l}^{j}$ and $r_{u}^{j}$ in Equation (9) denote the effective limiting side lengths of the rectangle, and $\theta_{l}^{i}$ and $\theta_{u}^{i}$ in Equation (10) denote the effectively perceived azimuth angles of the antenna limits. Figure 6 shows the position dependence.

The characteristic quantities related to the region average energy include moving average energy, average power ratio, and the region power correlation coefficient. The moving average power energy is related to the length of the moving average window of the previous frame, the average power ratio indicates the ratio of a region to the total energy, and the region power correlation coefficient indicates the correlation between different regions. To determine whether there is a valid human target in the detection target region, the energy-intensive motion characteristics of the region in a given frame are judged by the region. From the target detection, it can be seen that the presence of a valid target person has a relatively large impact on the energy intensity of the region. The average zonal energy of the *i*th region at frame *t* is expressed by the following equation:(11)$$Q_{i}\left[t\right]=\frac{1}{\left|Z_{i}\right|}\sum S_{n,m}\left[t\right]$$

#### 3.3.2. Personnel Saturation Detection

In the experiments, the radar transmitting component does not need to detect dangerous behaviors of people while the vehicle is parked, and detection of specific areas requires time and time constraints. Although the target area is a single in-vehicle detection location and there is no personnel exchange arrangement, there is still a decision result of setting two detection saturation states, i.e., occupied and unoccupied, in the experimental preprocessing stage, using “1” for occupied and “0” for unoccupied. The distribution state of drivers can be represented by the expression:(12)$$O_{i}\left[t\right]\in 0,1$$

$O_{N}$ represents the effective probability distribution, and the superposition of the effective area, $i^{x\left[t\right]}$, will obtain the actual effect of personnel saturation. In the experimental device, the definition of the spatial position probability distribution of the driver and passengers at the target position is expressed by:(13)$${\hat{p}}_{\left(O_{1},O_{2,\cdots ,}O_{N}\right)}\left[t\right]=g\sum_{i=0}^{N}W_{\left(O_{1},O_{2,\cdots ,}O_{N}\right)}i^{x\left[t\right]}$$

g(⋅) is a no/n-linear function, and the logistic regression functions is $g\left(z\right)=\frac{1}{1+e-z}$.

### 3.4. CNN Classification Models

#### 3.4.1. The Overall Process of Classification

Based on the comparative analysis of the above methods, the dataset of dangerous driving behaviors requires deep, large processing channels, and the degree of generalization of the model needs to be enhanced. Therefore, this paper uses the micro-Doppler sensitivity of the deep learning AlexNet framework, which can respond to larger values, to construct an identity verification method based on waveform number-mode analysis of FM continuous-wave echoes, while the traditional edgeless sensing technique is based on a priori learning of a large number of access points of subcarriers and fingerprint libraries. The 77 GHz millimeter-wave radar has an excellent angular resolution, which can obtain accurate Doppler mapping information. The highly integrated nature of the radar allows the use of small antenna arrays, and the centralized arrangement allows similar Doppler signals to be obtained at the same distance. The randomly distributed thermal noise signature is eliminated by the uncorrelated superposition of the radar’s parallel channels to obtain a thermally noisy Doppler signature over the same distance. The robustness of the driver’s motion is significantly improved, and high packaging saves development costs and time while also saving vehicle space. WiGig is a crucial technology for high-bandwidth, high-speed communication based on LANs and backbone networks. Processing signals using WiGig’s information transfer rate of up to gigabits allows effective extraction of real-time motion recognition features and from micro-Doppler elements rather than standard signal amplitude features, thus improving the accuracy of the features [25]. The results show that after parameter estimation and image preprocessing of the target signal, the mm-DSF dual-channel depth neural network with micro-Doppler features as input can increase the network depth and achieve effective classification [26,27].

The mm-DSF CNN used in this paper adopts the underlying system structure of AlexNet with good overfitting and uses ReLU as the activation function. The model consists of a two-channel convolutional network with six convolutional layers in one channel, two pooling layers, two fully connected layers, and finally, a SoftMax layer to output the classification results. Six convolutional layers are set up because the complexity of micro-Doppler requires increasing the depth of the convolutional layers. First, the redundant and spatially dense LRN layer is removed because the function of this layer is to make the large values of the response larger since small data parameters also take up storage and system learning time and have little impact on the classification. This operation is designed to improve the efficiency of the system. The ambiguous effect of average pooling of traditional convolutional neural networks is discarded, and the convolutional features with reduced dimensionality are represented by the maximum pooling operation. In the setting of the convolutional kernel size, since the dataset is the raw waveform signal of human action rather than image information, the size of the convolutional kernel is set to 3 × 3 to make the training process light and fast because the classification area of the feature distribution is relatively small. As shown in Figure 7, the convolutional layers are all the same and the pooling layers are all valid. Since the parameters of the convolutional kernel are small, to avoid overfitting, a convolutional layer is added in front of each pooling layer after feature extraction to increase the network depth and reconstruct and enhance the data structure. Using multidimensional convolutional neural networks for feature extraction of inter-frame continuity of target actions, the accuracy and convergence of action types can be greatly improved by layered horizontal fusion of data based on time series. The accuracy and convergence of action time can be greatly improved.

#### 3.4.2. Lateral Fusion Method Based on SlowFast CNN

To improve the recognition rate of action recognition and reduce the redundancy of feature data, the feature action information captured at reduced frequency is simultaneously input into two parallel convolutional neural networks, which are divided into two processing channels according to the timing drive. The low-frequency sampling channel captures the slow target signal and filters the reflected clutter. The fast-sampling network channel captures the larger waveform of the target waveform, so the channel is very lightweight. In this experiment, the data in the sampling rate are set to 120 frames per second, as this setting means capturing 6 frames per second.

The training samples are processed with a convolutional neural network. The network lateral fusion process is shown in Figure 8. The classification of dangerous driving behavior is specified by setting a uniform weight and a threshold that best fit the classification in the first step, with the threshold artificially specified in the range (−1, 1). The frame refresh rate of the low-frequency sampling channel is slow, capturing carrier information for the channel’s sparse frames, the high-frequency sampling channel operates with fast refresh and high temporal resolution, and the high-frequency channel operates with a tiny step of $\frac{\tau }{\alpha }$. α>1 is the ratio of the frame rate between the fast and slow paths, so the channel density of the high-speed path is α times that of the low-speed path, and the value of α set in this experiment is 8. According to the experimentally set channel capacity, the fast channel convolution kernel expression is $\left\{8T,S^{2},C\right\}$, the slow channel convolution kernel expression is $\left\{T,S^{2},4C\right\}$, the channel lateral fusion is unified in the format of dual channels, and the convolution kernel for dual-channel fusion operation is $\left\{5\times 1^{2},2C\right\}$.

In step 2, the sample set is denoted as $\left(x_{i},y_{i}\right)$, and the output bandwidth of the system network can be expressed as the following equation based on the expression of the sample set:(14)$${\hat{y}}_{j}^{i}=f\left(\beta_{j}-\theta_{j}\right)$$
where f(x) in the expression represents the ReLU activation function, $\theta_{j}$ represents the threshold of the jth neuron in the layer passed in the neural network, $\beta_{j}$ represents the real-time input value of the same neuron in the transport layer of the neural network, and $\omega_{ij}$ represents the weight between the output layer of the neural network and the *i*th and *j*th neurons of the hidden layer. Therefore, the expression for the input value is as follows: (15)$$\beta_{j}=\sum_{i=1}^{n}\omega_{ij}x_{i}$$

The third step is to evaluate the mean squared error of the neural network matrix $\left(x_{i},y_{i}\right)$. Since the output of the convolutional neural network is similar to multiple output edges due to the error, ${\hat{y}}_{j}^{i}$ denotes the actual output, and $y_{j}^{i}$ denotes the desired output value, the specific calculation procedure is improved to the following expression:(16)$$E=\frac{1}{2}{\sum_{j=1}^{m}\left({\hat{y}}_{j}^{i}-y_{j}^{i}\right)}^{2}$$

In step 4, the input neural network is conditioned on the minimum value of the error, bounded by the conditions allowed for learning, and with a threshold of the number of learning times. For everyday learning, the above conditions are satisfied, otherwise, the weights of mm-DSF CNN will be updated toward the bias of the target threshold, setting the learning rate of mm-DSF CNN to η. The model weight update is expressed as the following equation:(17)$$\Delta w_{ij}=-\eta \frac{\partial E}{\partial w_{ij}}=\eta {\hat{y}}_{j}^{i}\left(1-{\hat{y}}_{j}^{i}\right)\left(y_{j}^{i}-{\hat{y}}_{j}^{i}\right)x_{i}$$

The last step is to loop through the second, third, and fourth steps once the initialization is complete, and keep winding until the end of the conditions are not satisfied. At the end of the loop, the threshold and weights of the neural network need to be determined.

## 4. Experimental Design and Analysis

### 4.1. Experimental Design

The main equipment for the experiments in this paper is a millimeter-wave radar from Texas Instruments with a clock frequency of 77–81 GHz, a wavelength of about 4 mm, and a bandwidth of 4 GHz, which has a superior ultra-high frequency bandwidth in hardware compared to traditional wireless sensing devices and ensures highly accurate recognition of fine motions. The experimental equipment mainly includes the IWR1642 radar sensor device and the DCA1000EVM data acquisition card, the sensor holder, the Lenovo R7000 i5-7300HQ GTX1050 802.11AC wireless network card laptop, and the trimaran and SUV. For the experimental scenario, the radar sensor was fixed on the B-pillar on the driver’s side of the vehicle in clear weather at a height of 70 cm from the vertical height of the vehicle chassis, and the radar transmitter and receiver were placed spatially perpendicular to each other in the direction of the antenna facing the driver’s chest, placing the human body position exactly within the radar scan sector to obtain the complete behavioral characteristics of the target. The experiment consisted of six people of different weights and heights, distributed between 22 and 29 years of age and between 1.55 and 1.90 m tall. The sample-set for this experiment was a test set consisting of 9 actions, with 1 acquisition lasting 6 seconds, and a person’s specific target action was repeatedly tested and executed 50 times, for a total of 2700 sets of data information. The data samples for each data acquisition set were 255 × 255 × 3. By setting the Doppler sensing distance to 0.66 and 0.96 m to generate two 2D FFT velocity profiles at two different distances, the effective trajectory and physical characteristics of the target motion can be captured by accurately detecting the velocity of the target point at two fixed distances.

The number of Chirp samples per unit is set to 128, the number of Chirp signals per unit frame is 128, the fixed frame period is 30 ms, and in the case of each target driving action set to capture 200 frames, the capture time of each target action is 6.0 s according to the frame period, and the velocity resolution is 0.070 m/s. There is thermal noise in the daily air medium, and according to the Boltzmann constant and the effective noise bandwidth, the value of the maximum unambiguous velocity is ±4.564 m/s.

For the driving process, the relationship between risky driving behavior and driving psychology is inextricably linked, and Dula’s exploration of decision-making stipulates that variables of driving psychology have significant pattern changes for the production of risky driving behavior. Aggressiveness, anxiety, and comfort are the three major factors that contribute to the production of dangerous driving. This method was set up to experimentally conduct an integrated psychological questionnaire (DBQ) on 150 individuals with driving qualifications, including the amount of frustration, personality traits (TEIQue-SF) [28,29], environmental conditions, and traffic saturation, which had a direct effect on aggressive personality, and the Dula Danger Questionnaire (DDDI) for aggressive behavior and negative perceptions [30,31]. Based on the similarity of the behaviors, the actions were assessed in different dimensions, and the conditions for the establishment of the dangerous driving actions were determined by the comprehensiveness of each survey combined with the character personality and psychological environment influences, and this condition was used as the only criterion for the classification of the model identification. The data collection part of the experiment in this paper was conducted entirely outdoors to classify the typical driving hazard actions of drivers, so the object of identification is the driver’s single driving behavior. The single bad driving behaviors screened out mainly include: one-handed operation of the steering wheel (A), low picking up objects (B), violent movements (C), drinking water while driving (D), answering phone calls while driving (E), turning to the right after panning (F), turning 90 degrees to the right (G), turning to the left after panning (H), and turning 90 degrees to the left (I), covering 9 main movements. Subsequent sections will use the alphabetical symbols in place of specific actions.

In the detection of driver actions, the identity of the subject is not unique, including men and women of different heights and sizes. Variability, as a change in the identity label of the subject, must be set as constant and quantitative to capture specific waveform signals under different actions, and setting different vehicle speeds in comparison experiments has little or no effect on the recognition of target actions, so variability includes device type, data acquisition time, frame number, in the same scene, etc. During the recognition process, 50 samples of the same person were tested for the same action. Due to the limited space in the car, some limb actions could not be recognized due to artificial occlusion. In the case that the background is not very complicated, all 9 groups of standard actions can be recognized accurately in mm-DSF CNN, and the experiments show that a smaller data volume can be recognized effectively in mm-DSF CNN, and the actions are more targeted and better generalized.

To match the pictures to the pulse signal data, the naming of the pictures was standardized, for example, the first set of one-handed driving data was named bin_actionA_1.jpg, and the pictures taken of the driving actions were represented by the data acquisition transfer format bin_ and the letter of the action. The advantage of this set up is that the dataset can be organized later to avoid confusion of data image information and to reduce the hassle of manually labeling the information.

### 4.2. Experimental Analysis

#### 4.2.1. Model Performance Analysis

The network framework used in the collation of this experimental dataset uses the PyTorch model to build the overall system. Using compilation software with Python 2.7 and Anaconda, PyTorch can support Python 2.7 environment configuration, and PackageManager selects anaconda by expe rimental science and feasibility, requiring the installation of the powerful data science toolkit Anaconda. Ten samples from each group in the training set of the motion detection experiment were selected as test samples for the training set for the data network framework update. This experiment used a self-built dataset for experimental validation. The two-channel convolutional neural network mentioned above (including the low-speed sampling channel and the high-speed sampling channel) is shown in Figure 9. In the experiments to verify the effect of the separated channels, it can be seen that both separated channels have very regular characteristic change curves by way of fusion collaboration.

The results from the figure show that the loss rate of the low-frequency sampling channel is lower than that of the high-frequency sampling channel when capturing the slow-motion micro-motion during the sampling signal, indicating that the low-frequency sampling channel produces fewer failed samples during the classification process. However, the sampling accuracy of the low-frequency sampling channel for the signal is not as good as that of the high-frequency sampling rate, indicating that the data sample set of dense and high frequency has a good recognition base with a highly accurate recognition rate. Moreover, both the loss rate and the accuracy rate level off as the iteration cycle proceeds. Therefore, changing the input channel to dual-GPU, dual-channel processing can improve the performance of the experimental model.

#### 4.2.2. Combined Eigenvector Analysis

Six eigenvectors can be extracted from the collected micro-Doppler spectrogram, and the spectrogram consists of these six important eigenvectors. Different micro-Doppler spectrograms can be represented by the classification of the different values of the eigenvectors. Due to the different methods of target motion, there exist feature vectors with large and small impacts on the micro-Doppler spectrum classification. The feature vectors with a very small impact are removed based on the feedback from the classification effector, thus avoiding overfitting during the classification process. The 6 feature vectors can be freely combined to produce up to 720 feature combinations, and the accurate analysis of the classification effects of the feature vectors and feature combinations yields that the feature vectors torso micro-Doppler vector, $v_{torso}$, shoulder micro-Doppler vector, $v_{shoulder}$, and arm micro-Doppler vector, $v_{\lim b}$, have a strong impact. The experiments showed that the feature vectors $U_{1}$, $U_{2}$, and $U_{3}$ were much less effective than the first three in terms of results. Especially, E and F motion have almost no effect on the classification results of dangerous driving. Based on the above evaluation of the feature vectors, it was necessary to select the better combination of feature vectors. This included a combination of three feature vectors: torso micro-Doppler vector, $v_{torso}$, shoulder micro-Doppler vector, $v_{shoulder}$, and arm micro-Doppler vector, $v_{\lim b}$.

As shown in Figure 10, the nine hazardous actions in the dataset were accurately identified, and the classification effect accuracy was divided into test accuracy for known persons and test accuracy for unknown persons. The confusion matrix provides a visual representation of the accuracy distribution of each feature and also represents the classification results and the actual measurements. Each row represents the test type and each column represents the attribution type. The confusion matrix is characterized by a simple concentration of values on the main diagonal, with the horizontal and vertical axes representing the classification results and the true data type of the test dataset, respectively. The confusion matrix for each action tested by known persons shows that a maximum accuracy of 97% can be achieved for action A (one-handed steering wheel), and the confusion matrix for each action tested by unknown persons shows that an accuracy of 96% can be achieved for action H (left side horizontal and turn back). The minimum test accuracy is no less than 93%, so the individual algorithms of this model can achieve the accuracy needed to detect dangerous actions, both for known and unknown persons.

#### 4.2.3. Analysis of the Impact of Unrelated Targets in the Vehicle

This experiment is unrelated to directionality since the radar fixation is directly on the driver’s effective energy coverage area. The effect of personnel irrelevancy is the effect of the passenger seat in the car. The irrelevant target person in the back row’s target behavior recognition was analyzed, so four sets of comparison experiments were set up: no interference, interference from the passenger side, interference from the back row, and interference from the passenger side. As seen from the figure below, the passenger seat and back row have no significant effect on the classification of dangerous movements of the target because the millimeter-wave radar produces different spectral maps for directions at different distances and is highly directional. The effective distance for target identification was set from 1.1 to 1.75 m. The distance for the passenger seat is smaller than the distance for detecting the target, and the distance for the back row is larger than the distance for detecting the target, so the interference of extraneous targets can be effectively excluded. The effect of target interference is shown in Figure 11.

#### 4.2.4. Analysis of the Diversity of Personnel

In the experimental set up for six subjects with significant differences in age, height, and weight, the issue of the effect of individual driving habits and target height and weight on the effective recognition distance of the radar receiver resulted in weak differences in the micro-Doppler spectrum of the input training set for the same maneuver for each subject. Figure 12 shows the box line plots for the six issues.

With the two-channel network model, each subject showed small variability for each of the nine actions in the experimental set up. Subject b had a small error in accuracy under the same conditions, with a maximum precision of 96.8% and a minimum accuracy of 95.8%. Subject c could achieve a maximum accuracy of 87% and a minimum of 85.9% for the same action. Both were able to satisfy the accurate classification of dangerous driving actions with little difference in the accuracy of the same scheme. In the random action data statistics for the same target, the float in accuracy is greater than the same action accuracy, but the normalized data illustration still demonstrates the problem of the generalizability of personnel differences, indicating the high robustness of the model’s approach.

#### 4.2.5. Sample Size Optimization Analysis

To determine the effect of sample size on the accuracy of action recognition, experiments were conducted on test sets of 10, 50, and 100 oversampled datasets for each activity under the same environmental and personnel conditions. The accuracy of the oversampled 100-group sample was found to be higher than that of the 10-group and slightly higher than that of the 50-group training. Since 10 sets of training samples are not enough, and 100 can achieve the expected results, increasing the training Doppler spectrum appropriately can effectively improve the recognition accuracy. As the driving system is time-sensitive, continuing to increase the number of samples will increase channel density, which will increase the time complexity, thus increasing the warning delay. Figure 13a reflects the relationship between the number of samples and recognition accuracy. Figure 13b shows a shortened iteration period and increased aggregation speed for data streams with a large sample size in the same personnel environment.

#### 4.2.6. Analysis of Identification Results

The hazardous action recognition method in this paper is a two-channel convolutional neural network based on micro-Doppler features. In vector features, the spectrograms of tiny action features of the target actions can be extracted, and the coarse- and fine-granularity hierarchical optimization classification can be performed according to the micro-Doppler spectrograms, which fully meet the standard requirements of driving warning. The common method uses CNN single-feature time–distance feature classification [32,33], which has a significant loss of classification accuracy after putting the Doppler feature vector dataset into this model, and the classifier cannot verify the classification of head motion and shoulder micro-Doppler vectors due to the slight distance difference, losing a part of the accuracy. The second general approach uses ST-GCN to classify critical points of the human skeleton [34,35], which can achieve a classification rate of 89.4% using a graphical convolutional network due to the significant differences in the physical signs and postures of the participants. The conclusion is displayed in the figure below, where the method of this paper classifies dangerous driving maneuvers, the technique, and results slightly better than the other two effects. Figure 14 compares the experimental results when using different methods for the same dataset.

## 5. Conclusions

In this paper, we proposed a new integrated recognition technique for recognizing dangerous driving maneuvers based on FWCW millimeter waves by adjusting the radar parameters and updating the classification method. A mm-DSF network was used in the Doppler signal feature vector representation, and convolutional layers were added before each pooling layer to increase the network depth to achieve more accurate hazardous driving classification results. The signals from the neurons were input to the synapses, selective channel processing was performed according to the SlowFast CNN dual-channel for different sparsity of the feature vectors, and the high-frequency sampling channel was simultaneously fused with the low-frequency sampling channel for lateral data fusion to achieve reliable data results. The linear input FM continuous wave signal after feature extraction was separately trained and tested. The experimental results show that the robustness and accuracy of the dual-channel micro-Doppler features of this paper with CNN single-feature time–distance features and ST-GCN are better for the critical points of the human skeleton. For the dangerous driving behavior of six training objects, the accuracy of upper body recognition can reach 96.9%, and the lowest recognition accuracy for training objects was 85.6%. The overall average recognition accuracy of this method is 1.97% higher than the accuracy of traditional techniques, so it proves the model’s effectiveness and has a strong generalization and inclusion for the mass population. The method will be improved in the future to enhance the accuracy and identify the risky behaviors of multiple people in the car.

## Figures and Tables

**Figure 1 sensors-22-08929-f001:**
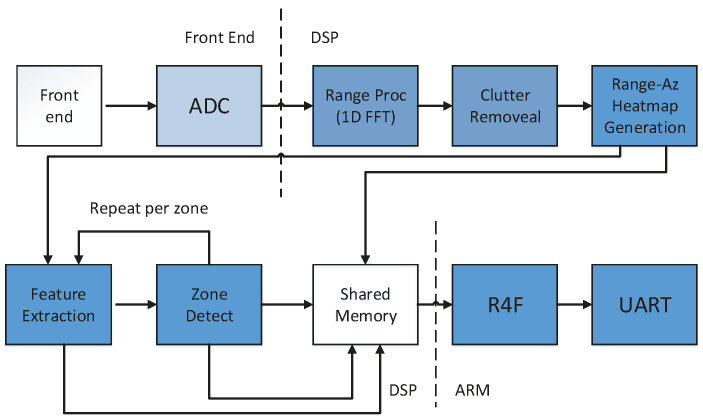
Radar handling process.

**Figure 2 sensors-22-08929-f002:**
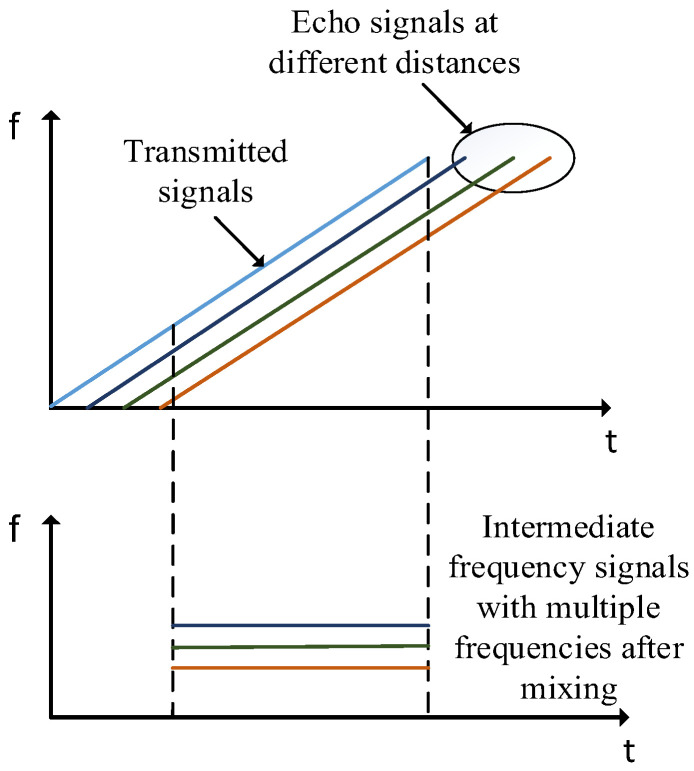
IF signal generation process.

**Figure 3 sensors-22-08929-f003:**
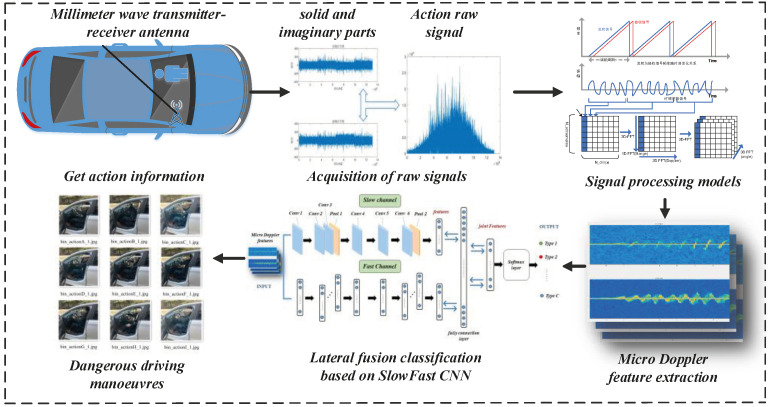
Overall flow chart of the mm-DSF model.

**Figure 4 sensors-22-08929-f004:**
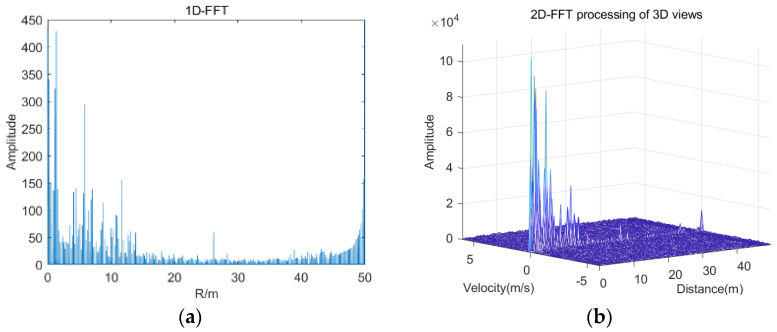
(**a**) Distance Doppler spectrogram (bending down to pick something up). (**b**) Distance-velocity Doppler spectrum (bending down to pick something up).

**Figure 5 sensors-22-08929-f005:**
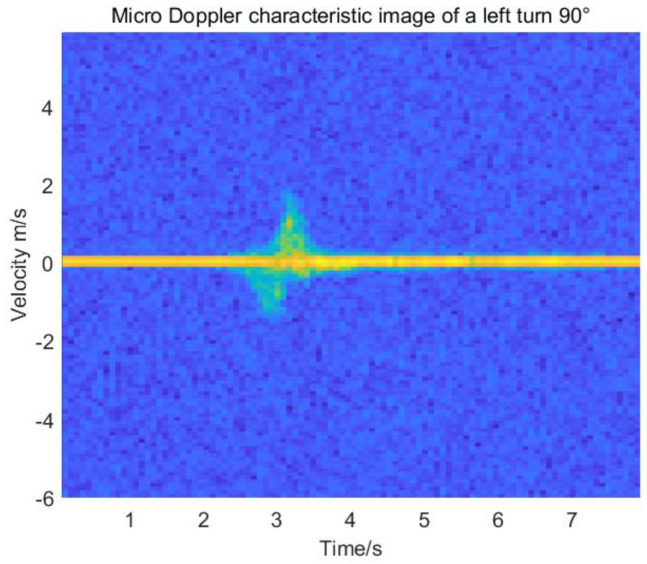
90° micro-Doppler characteristic of the left rotating body.

**Figure 6 sensors-22-08929-f006:**
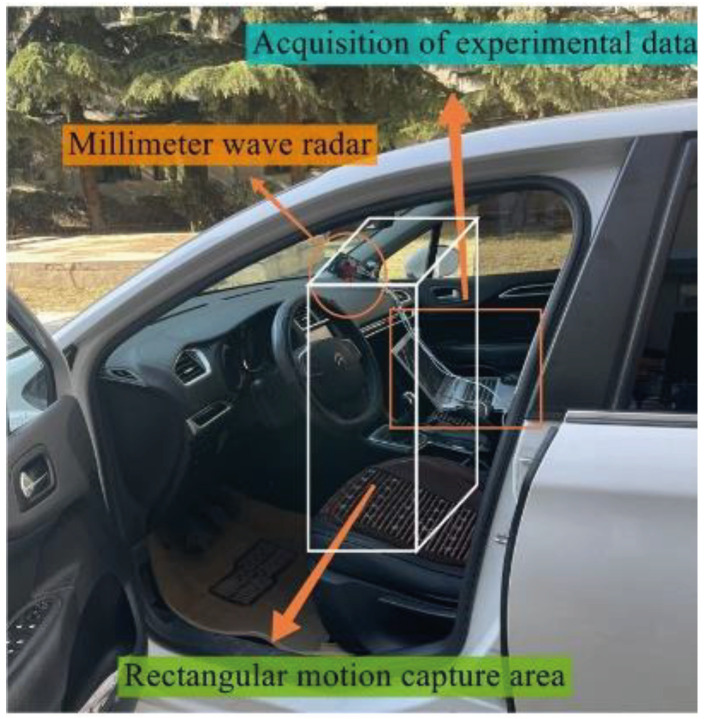
Radar effectively identifies equipment and space.

**Figure 7 sensors-22-08929-f007:**
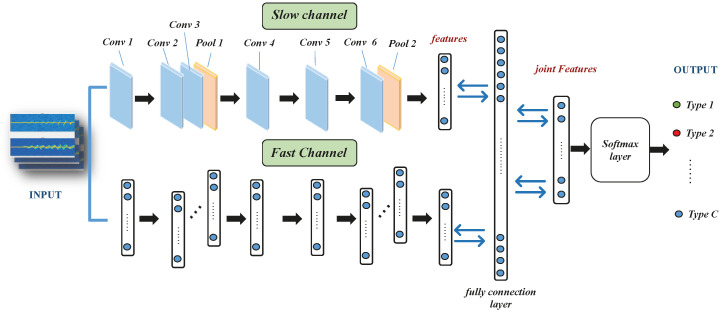
CNN classification flow chart.

**Figure 8 sensors-22-08929-f008:**
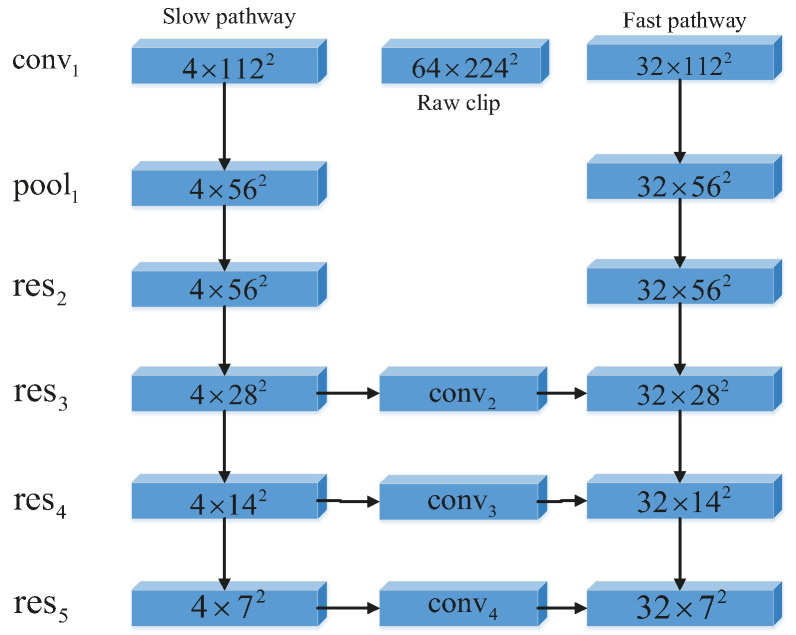
Flowchart of SlowFast CNN for lateral fusion.

**Figure 9 sensors-22-08929-f009:**
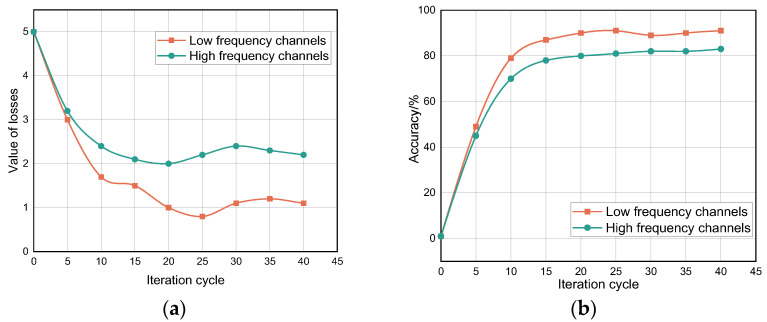
(**a**) Variation in sample loss rate. (**b**) Variation in sample accuracy rate.

**Figure 10 sensors-22-08929-f010:**
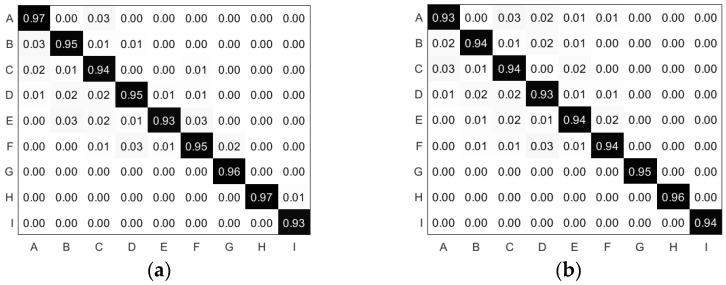
(**a**) Confusion matrix for each action test of the test for known persons. (**b**) Confusion matrix for each action test of the test for unknown persons.

**Figure 11 sensors-22-08929-f011:**
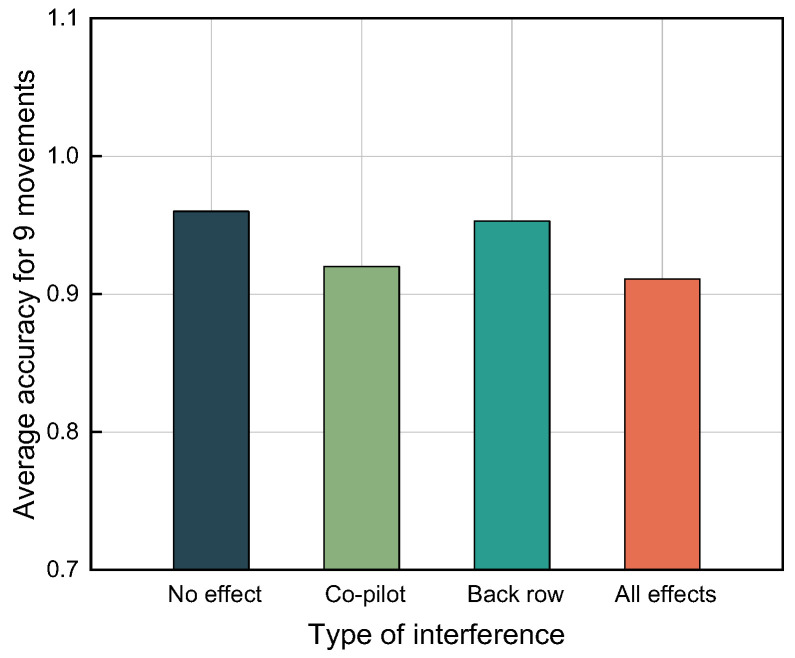
Histogram of the accuracy of unrelated target interference.

**Figure 12 sensors-22-08929-f012:**
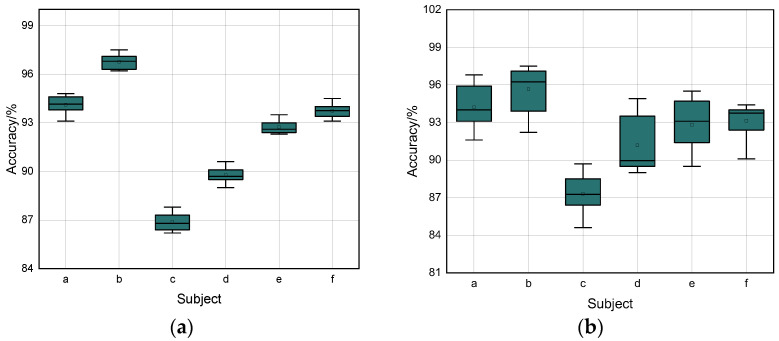
(**a**) Box line plot of 6 subjects for the same action. (**b**) Box line plot of 6 subjects for random actions.

**Figure 13 sensors-22-08929-f013:**
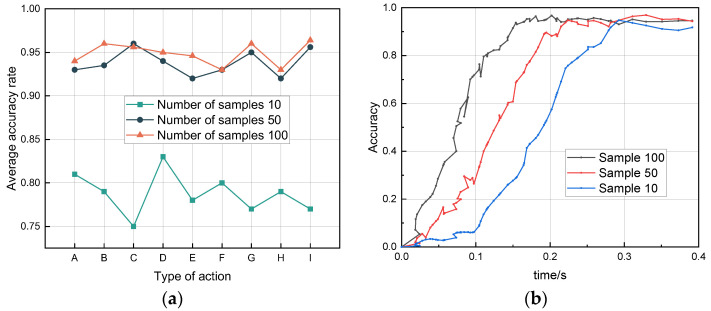
(**a**) Sample size analysis line graph. (**b**) CDF plot of iteration speed with different numbers of samples.

**Figure 14 sensors-22-08929-f014:**
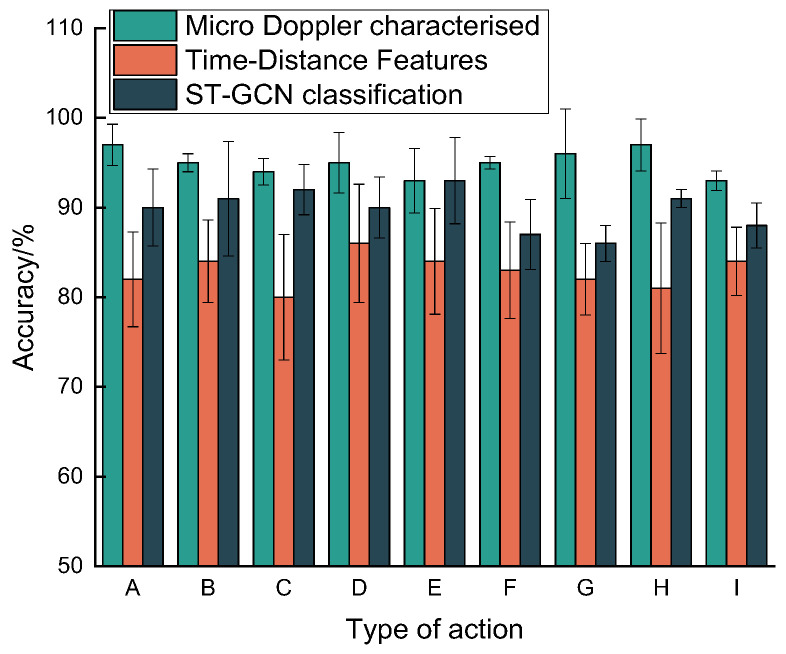
Histogram for analysis of experimental results.

## Data Availability

Not applicable.

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
