# Peer review of "mm-DSF: A Method for Identifying Dangerous Driving Behaviors Based on the Lateral Fusion of Micro-Doppler Features Combined"

_sensors, 2022, doi:10.3390/s22228929_

Round 1
Reviewer 1 Report
The work proposed a method to identify the dangerous driving behaviors, which is of research significance. I have certain concerns:
1. The last sentence of the first paragraph in Introduction section, "The judgment criteria of dangerous driving behavior used in this paper are judged by a large number of emotional intelligence questionnaires and Dula danger questionnaires (DDDI), in the analysis of character and human personality habits, to obtain multidimensional results of psychological adaptation to the environment, integrated by the hand masking algorithm, and analyze the typical dangerous driving behavior for classification and identification." should be placed at the experimental description part.
2. The second and third paragraph of the Introduction should be rewritten as one paragraph.
3. The Introduction section and the Related work section should be unified as on section.
4. The description of d of equation(1)is not correct. It stands for differential operator not distance.
5. There does not exist Δtf in equation(2), why do the authors include a description for it.
6. Please reformat equation(11)-(15). They show excessive larger font.
7. The authors should provide a clean version of Figure 7.
Reviewer 2 Report
In this contribution, a method for identifying dangerous driving behaviors based on the micro-Doppler features is reported. The article is well-written except for some sections, as reported hereafter.
The English level is good and the topic appropriate for the present journal.
I consider very important for a such an article, introducing an accurate comparison with the scientific literature, highlighting the novelty of the paper. This should be a mandatory requirement for every contribution. In detail:
I consider the Introduction, particularly the first part, too verbose. I suggest shortening this section.
Also the section 3 should be shortened, by maintaining only the most advanced information, e.g., those related to the phase analysis, as in:
E. Cardillo et al., “Embedded heating, ventilation, and air conditioning control systems: from traditional technologies towards radar advanced sensing,” Review of Scientific Instruments, vol. 92, Issue 6, 061501, pp. 1-14, Jun. 2021.
But in the literature you can find a lot of contributions, also from the same research group of the last reference.
There are a lot of contributions on this topic, you should highlight what are the novel elements of your contribution. A comparison with the state of the art is required.
As an example, you can find interesting papers exploiting similar techniques aimed at exploiting millimeter-wave radar sensor for the detection of child presence for automotive anti-abandon systems.
Author Response
Dear Reviewer,
Thank you very much for reviewing our paper, and for your comments and detailed suggestions that helped us to improve our paper.
Please see attached file.

Round 2
Reviewer 2 Report
The authors addressed all my concerns.